# Extra-Thyroidal Impacts of Serum Iodine Concentrations During Early Pregnancy on Metabolic Profiles and Pregnancy Outcomes: Prospective Study Based on Huizhou Mother–Infant Cohort

**DOI:** 10.3390/nu17101626

**Published:** 2025-05-09

**Authors:** Zhaomin Liu, Chaogang Chen, Cheng Wang, Yaqian Wang, Minmin Li, Wenjing Pan

**Affiliations:** 1Guangdong Provincial Key Laboratory of Food, Nutrition and Health, Department of Nutrition, School of Public Health, Sun Yat-sen University (North Campus), Guangzhou 510080, China; limm9@mail2.sysu.edu.cn; 2Department of Clinical Nutrition, Sun Yat-sen Memorial Hospital, Sun Yat-sen University, Guangzhou 510012, China; chenchg@mail.sysu.edu.cn (C.C.); wangcheng@mail.sysu.edu.cn (C.W.); 3Huizhou First Maternal and Child Health Care Hospital, Huizhou 516000, China; wangyq229@mail3.sysu.edu.cn

**Keywords:** serum iodine concentrations, gestational metabolic syndromes, metabolic biomarkers, pregnancy outcomes

## Abstract

**Objectives:** This study aimed to test the extra-thyroidal impacts of maternal serum iodine concentrations (SICs) on metabolic factors and subsequent pregnancy outcomes. **Methods:** Single pregnant women aged 18–49 years were recruited during their first prenatal visits. SICs at first trimester (T1) were tested by ICP-MS. Metabolic factors [body mass index (BMI), fat %, glucose, lipids, uric acid, and blood pressure] were measured, and composite indices [the triglyceride–glucose (TyG) index, TyG-BMI, and the Framingham steatosis index (FSI)] were estimated. Obstetric and birth outcomes were retrieved from the hospital information system, including gestational diabetes (GDM), gestational hypertension (GH), fetal distress, postpartum hemorrhage, premature rupture of membrane, small and large for gestational age (SGA and LGA), preterm birth, and low birth weight. Multivariable linear and logistic regression models were applied to explore the associations between maternal SIC, metabolic factors, and pregnancy outcomes. **Results:** A total of 1456 mothers were included for analysis. Maternal LgSIC values at T1 were inversely associated with early gestational weight gain (*β* = −0.113, *p* < 0.001) and BMI at T1 (*β* = −0.070, *p* = 0.006), but they were positively associated with triglycerides (*β* = 0.142, *p* < 0.001), the TyG index (*β* = 0.137, *p* < 0.001), and uric acid (*β* = 0.060, *p* = 0.018). However, upon further adjustment for thyroid hormones, the associations were attenuated. The joint effects of high SIC and metabolic conditions (hyperlipidemia, high FSI, and GH) suggested increased adverse pregnancy outcomes (increased postpartum bleeding, reduced birth length, and reduced delivery weeks). **Conclusions:** Our prospective data in the iodine replete region indicated that high SICs at T1 were associated with increased risk of metabolic conditions and adverse birth outcomes, with the associations being independent of thyroid hormones.

## 1. Introduction

Gestational metabolic syndromes (GMSs), or metabolic abnormalities, have been associated with an increased risk of obstetric complications and adverse pregnant outcomes [1]. The etiologies of GMSs comprise multiple interactions between genetic and environmental factors [2]. Among them, diet plays an essential role in the process. Therefore, nutritional factors have received considerable attention for the prevention and control of metabolic disorders during pregnancy.

Iodine is an indispensable trace element required for the biosynthesis of thyroid hormones, which play important roles in various metabolic processes of mammals [3]. Iodine requirements during pregnancy are substantially increased due to elevated urinary iodine losses, increased thyroid hormone production, and iodine transfer via the placenta to the developing fetus [4]. Pregnant women are susceptible to both iodine deficiency and excess, especially in iodine-sufficient areas due to the high consumption of iodine-rich foods (dairy products and seafoods) and due to routinely prescribed iodine or iodine-containing supplements during gestation [5]. Even a slightly increased iodine load was reported to result in altered maternal thyroid function, which might irreversibly impact fetal growth and development [4]. Recent studies suggest that iodine, aside from its role in the thyroid, functions as an antioxidant, immunomodulator, and differentiator in various organs and tissues [6,7]. This is because iodine-specific transporters (i.e., sodium/iodide symporter and PENDRIN) are expressed not only in thyroid follicular epithelial cells but also in many other organs, such as the salivary glands, stomach, placenta, lactating mammary gland, ovary, prostate, and pancreas [6]. In vitro and animal experiments showed that oxidized iodine can neutralize reactive oxygen species and trigger apoptosis, and it has differentiation effects on diverse epithelial cells [6]. Moreover, iodine can be taken up and metabolized by immune cells, acting as an anti-inflammatory or proinflammatory agent [7]. However, few studies have examined the extra-thyroidal effects of iodine (independent of thyroid hormones) during gestation.

Observational studies on the associations of iodine with metabolic conditions or components have reported controversial findings [3,8], with limited studies being conducted among pregnant women. The disparities might be due to the utility of different indicators for assessing iodine nutrition (urinary iodine, water and salt iodine, dietary iodine, or iodine supplements), different study designs, and different population features. The majority of previous studies applied the urinary iodine concentration (UIC) for the assessment of personal iodine status. Although recommended by the WHO for appraising populational iodine status, the UIC, especially the spot UIC, had notable disadvantages in reflecting individual iodine status due to its high variabilities with drinking water, consumed foods, circadian rhythm, and seasonal changes [9,10]. The serum iodine concentration (SIC) has recently been indicated as a valid biomarker for evaluating personal iodine status because it is relatively stable and reflects the bioactive iodine component [9,10]. However, observational studies utilizing the SIC as an individualized iodine marker are more limited.

Most previous studies on iodine and metabolic factors were conducted in iodine-insufficient regions, while limited studies have been conducted in iodine-replete regions (i.e., seaside cities). With the introduction of the universal salt iodization policy in China since the 1990s, there have been increasing concerns that an excessive iodine intake might have adverse impacts on human health, including metabolic disorders, especially in iodine-sufficient regions and among vulnerable subpopulations [10,11]. Fetus and neonates are at particular risk for developing hypothyroidism if exposed to an iodine excess in utero, as the fetal thyroid remains immature until near term, placing them at risk for failure to escape from the acute Wolff–Chaikoff effect [12]. Until now, no study has explored the joint impacts of maternal iodine and metabolic conditions on pregnancy outcomes.

We therefore aimed to examine the associations of gestational SIC and metabolic factors (maternal weight change, glucose and lipid metabolism, uric acid, blood pressure, and insulin resistance) during early pregnancy with subsequent obstetric complications and fetal growth abnormalities in an iodine-replete region (Huizhou, a coastal city in south China). The findings can help in elucidating the extra-thyroidal impacts of iodine on gestational metabolic components and subsequent birth outcomes.

## 2. Materials and Methods

### 2.1. Participants and Recruitment

This study was conducted within the Huizhou mother–infant cohort, which was a joint project conducted by the School of Public Health of Sun Yat-sen University (SYSU) and Huizhou 1st Mother and Child Health Care Hospital (HMCHH). Ethical approval was obtained from the ethics committee of HMCHH (registration no.: 2018002). All the participants signed written consent forms before enrollment. The project has been registered in ClinicalTrials.gov (NCT03922087).

A flow chart of this study is shown in Figure 1. Single pregnant women aged from 18 to 49 years and who had been a regular resident in Huizhou city for at least 5 years were enrolled during their first antenatal visits. From December 2018 to August 2021, a total of 4848 pregnant mothers were recruited. Among them, 1558 were selected for testing serum iodine concentrations (SICs) in their early pregnancy. For a detailed outline of the study protocol, refer to our prior publications [9,13]. Mothers with a diagnosis of pre-pregnancy diabetes or thyroid disorders, undertaking treatment for thyroid dysfunction, with an intake of iodine-containing drugs, prior or current hepatic disorders, including intrahepatic cholestasis during pregnancy, or with active virus hepatitis were excluded from analysis. A total of 1456 participants thus remained for the current analysis.

### 2.2. Data Collection from Questionnaire Survey and Biochemical Testing

The participants were asked to complete a questionnaire survey and to provide anthropometric measurements, and they were asked to donate fasting blood samples for biochemical testing during their first prenatal visits. Information on socio-demographics, medical and family history, medications, and lifestyle factors (dietary habits, smoking, tea and alcohol consumption, physical activities) were collected using a structured and pretested questionnaire, which was administered by trained research staff through face-to-face interviews. Maternal weight and height were measured in early pregnancy and before delivery by standard protocols. Pre-pregnant body weight was self-reported by the participants using their most recent measurements before gestation. Body mass index (BMI) was estimated by dividing body weight (kg) by the squared height (m^2^). Body fat was tested by a multi-frequency bio-impedance approach (NQA-PI, Beijing, China). The women were asked to empty their bladder and not to undertake intensive exercise or drink a lot of water before measurement.

After overnight fasting (8–12 h), venous samples were collected in either EDTA or blank tubes for the biochemical tests. Centrifugation was carried out at 3000 r/min for 10 min at 4 °C or below, and separation was carried out within 2 h after collection. In order to reduce the burden of blood collection from the participants, we used residual samples for SIC testing, which were collected soon after the routine clinical biochemical tests. The sample size for the SIC measurements was estimated with consideration of both the study’s purpose and the funding budget. To avoid iodine contamination, alcohol was used instead of iodophor disinfection for blood collection, and all the investigators were required to wear masks to prevent possible iodine contamination from saliva. Maternal SICs were tested by inductively coupled plasma mass spectrometry (ICP-MS, Agilent 7700x, Agilent Technologies, Inc., Santa Clara, CA, USA) according to the Chinese National Hygiene Standards of WS/T 783-2021 [14]. Iodine standards (GSB 04-2834-2011, ClinChek^®^ Serum Control, RECIPE, Munich, Germany) for quality control were tested intermittently for every batch of 20 samples. Plasma fasting and postprandial glucose (PG), uric acid (UA), and lipid (triglycerides and total cholesterol, TGs and TC) levels were tested by enzymatic colorimetry. Thyroid hormones, including free thyroxine (FT4), free triiodothyronine (FT3), and thyroid-stimulating hormone (TSH), in early pregnancy were tested by an electro-chemiluminescence assay. Except for SICs, all the other biochemical markers were tested in a Cobas Elesys 602 autoanalyzer (Roche Diagnostics, Basel, Switzerland), and the results were retrieved from the hospital information system. All the intra- and inter-assay coefficients of variation (CVs) were less than 5%.

### 2.3. Diagnosis of Gestational Complications and Metabolic Conditions

Gestational hypertension (GH) was diagnosed by the clinicians if BP > 140/90 mmHg after 20 gestational weeks (GWs). GDM was diagnosed by a standard 75 g oral glucose tolerance test during 24–28 GWs according to the recommendation of the International Association of Diabetes and Pregnancy Study Groups (IADPSG), i.e., if one or more of the following glucose levels were above the cutoffs: 0 h ≥ 5.1 mmol/L, 1 h ≥ 10.0 mmol/L, or 2 h ≥ 8.5 mmol/L [15]. GMS was determined if there was the presence of at least three of the following conditions: early pregnancy BMI ≥ 23.0 kg/m^2^, current GDM, current GH, or hypertriglyceridemia (TGs ≥ 1.7 mmol/L). GMS was defined with adaption to data availability (as we lacked data for waist circumference and HDL-c due to limited resources), gestational stage (as the samples were mostly collected during early pregnancy), the characteristics of the Chinese physique (BMI cutoffs for overweight or obesity) [16], and with reference to the WHO/IDF definition for adult metabolic syndromes. We defined hypertriglyceridemia as a plasma TG level above 1.7 mmol/L due to no remarkable increase in lipids in the first trimester of pregnancy [17]. The Framingham steatosis index (FSI) was estimated on the basis of metabolic conditions and liver function with the inclusion of parameters such as age, BMI, TGs, hypertension, diabetes, and ALT/AST. As a surrogate marker for non-alcoholic fatty liver disease (NAFLD), the FSI has been validated in Chinese population [18,19]. The FSI formula was modified due to its application in gestational populations by replacing hypertension and diabetes with GH and GDM. A detailed equation is indicated in the footnotes of the following table. The triglyceride–glucose (TyG) index is a reliable and simple surrogate for insulin resistance that is substantially relevant to metabolic disorders [20]. The TyG index was calculated as follows: In[TGs (mg/dL) × FPG (mg/dL)/2].

### 2.4. Obstetric and Birth Outcomes

We examined the following obstetric and birth outcomes: fetal distress, postpartum hemorrhage, premature rupture of membrane (PROM), birth weight and length, small and large for gestational age (SGA and LGA), preterm birth (PTB), and low birth weight (LBW). All the diagnostic criteria for obstetric complications were in accordance with the Chinese textbook of Obstetrics and Gynecology [21]. Fetal distress during labor was diagnosed by an experienced clinical gynecologist according to fetal heart rate and movement, pollution of amyloid fluid, fetal scalp blood tests, and Apgar scores after delivery. Postpartum bleeding volumes were estimated within 24 h after delivery. PROM was diagnosed as an amniotic membrane rupture at term before labor started. The adverse birth outcomes, LGA and SGA, were diagnosed as a birth weight higher than the 90% or lower than the 10% centiles compared to those of healthy ones of the same gender and gestational age, according to the 2020 Chinese Standard Growth Chart for newborns issued by the Capital Pediatric Research Center [22]. Low birth weight was defined as a birth weight less than 2500 g at term delivery. Preterm labor was defined as delivery before 37 completed weeks. All the clinical information and outcomes were retrieved from the hospital information system.

### 2.5. Statistical Analysis

Data entry was carried out using Epi-data 3.1. The data processing and regression analysis were performed using SPSS 24.0 and R 4.3.0. The Storm Statistical Platform (www.medsta.cn/software, accessed on 7 November 2024) was applied for the restricted cubic spline (RCS) analysis. Data normality was visually tested by a QQ plot. Variables with a skewed distribution or heterogenous variance (SIC, body fat %, TGs, TSH, FT3/FT4) were logarithmically transformed before the parametric approaches. Baseline characteristics of the participants (Table 1) were described by a different severity (0, 1–2, 3–4 scores) of GMS. Multivariable linear regression was applied to explore the associations of LgSIC in early pregnancy (T1) with various metabolic variables, including BMI at T1, early gestational weight gain (GWG at T1) and body fat % at T1, fasting and post-load glucose levels, HbA1c, TC, TGs, UA, and composite indices such as the TyG index, TyG-BMI, and the FSI (Table 2). Data on body fat % and HbA1c were only available for 709 and 188 women, respectively. Confounders in the regression models were selected according to the univariate results, the literature, or biological explanations. Further adjustment was made for thyroid hormones (LgTSH and LgFT3/FT4) in model 2 to testify the extra-thyroidal impacts of the SIC on metabolic factors. Collinearity tests were carried out to avoid possible bias in the regression models. The variables that are included in the composite formulas of the TyG index, TyG-BMI, and the FSI were not adjusted again in the regression models to avoid over-controlling or possible collinearity. Multivariable RCS models were fitted for testing the non-linearity between SIC at T1 and metabolic markers, with four nodes at the conventional centiles of 0, 35%, 65%, and 95%. Their relationship is visualized in the figure in Section 3. Multivariable piece-wised linear regression by high and low levels of SIC was further conducted (Appendix A) if non-linearity (*p* for linearity < 0.1) was suggested.

A multivariable general linear model (GLM) was applied to compare the estimated means of the metabolic variables by high vs. low levels (median of 88.9 μg/L) of SICs at T1 (Table 3). The joint effects of maternal SICs at T1 with metabolic conditions on pregnancy outcomes were testified by GLM analysis (Table 4) for continuous outcomes (postpartum hemorrhage volume, delivery weeks, birth weight, and birth length) or logistic regression (Table 5) for categorical outcomes (fetal distress, PTB, PROM, SGA, LGA, and LBW). The examined metabolic conditions included a high risk of GMS (≥1 items), overweight/obesity (pre-pregnancy BMI ≥ 23 kg/m^2^), GDM, GH, hypertriglyceridemia (TGs ≥ 1.7 mmol/L), insulin resistance (TyG index ≥ 8.73), and hyperuricemia (UA ≥ 357 μmol/L).

Sensitivity analyses were conducted to test the robustness of the results by comparing SICs at T1 by high vs. low levels of metabolic factors using GLM analysis (Appendix A) and by the exclusion of mothers with clinical/subclinical thyroid dysfunction (euthyroid, n = 1307), GH (non-GH, n = 1436), overweight/obesity (pre-pregnancy BMI < 23 kg/m^2^, n = 1337), or hyperlipidemia (normal TC and TGs, n = 1152) by multivariable linear regression (Appendix A). Subgroup analyses (Appendix A) were performed to explore if the associations differed among the various subgroups (i.e., GDM status, prim-parity or not, high and low TyG index). Interactions were tested beforehand using a univariate GLM or logistic regression, and the results of the stratification were reported for only those with a *p* of interaction less than 0.1.

## 3. Results

The participants’ basic characteristics showed that (Table 1) mothers with GMS (3–4 GMS items, n = 55) had a greater age and pre-pregnancy BMI, less educational attainment, and were more likely to have multi-parity and a medical history of GDM and GH than those without GMS. Mothers with GMS had a greater metabolic risk, including a higher prevalence of GDM and GH, a higher GWG and body fat % at T1, higher levels of TC, TG, HbA1c and UA, a higher TyG index and FSI but a lower FT3/FT4 ratio than mothers without GMS. Mothers with GMS were less likely to give birth vaginally and had a relatively short pregnancy duration.
nutrients-17-01626-t001_Table 1Table 1Basic characteristics of pregnant women who had their serum iodine concentrations tested (SIC) during early pregnancy by the number of items of gestational metabolic syndromes (GMSs), Huizhou mother–infant cohort (n = 1456).
Number of Gestational Metabolic Syndromes*p*
0 (n = 785)1–2 (n = 547)3–4 (n = 55)Maternal age, years27.7 ± 3.528.6 ± 3.732.1 ± 3.9<0.001Pre-pregnancy BMI, kg/m^2^19.4 ± 1.823.1 ± 3.926.5 ± 3.6<0.001Education, university or above, n (%)195 (24.9%)117 (21.5%)10 (18.5%)0.105Nulliparous, n (%)469 (59.9%)271 (49.5%)19 (34.5%)<0.001Smoking (active or passive), n (%)427 (54.4%)294 (53.7%)33 (60.0%)0.674Alcohol drinking, n (%)22 (2.8%)16 (2.9%)1 (1.8%)0.911Dietary seaweed and kelp intake, g/d (n = 157)10 (2.0, 22.5)10 (2.0, 40.0)20 (2.0, 80.0)0.811Usage of iodinized salt, n (%) (n = 729)176 (43.9%)148 (49.5%)16 (55.2%)0.534Gestationl weeks for biochemical testing, wks12.4 (12.0, 12.4)12.4 (12.1, 12.7)12.4 (12.0, 12.7)0.826Family history of diabetes, n%50 (6.4%)56 (10.2%)13 (23.6%)<0.001Medical history of GDM, n (%)13 (1.7%)43 (7.9%)8 (14.5%)<0.001Medical history of GH, n (%)2 (0.3%)8 (1.5%)2 (3.6%)0.028Medical history of PCOS, n (%)13 (1.7%)29 (5.3%)4 (7.3%)<0.001Current GDM, n (%)0186 (34.0%)49 (89.1%)<0.001Current GH, n (%)016 (2.9%)12 (21.8%)<0.001Current hyperlipidemia, n%0201 (36.7%)53 (96.4%)<0.001Current euthyroid, n%693 (90.5%)496 (93.2%)49 (90.7%)0.207GWG at T1, kg0.41 ± 2.201.39 ± 2.933.56 ± 3.97<0.001Body fat % (n = 709)26.2 ± 5.032.9 ± 6.335.7 ± 4.2<0.001Triglycerides, mmol/L1.04 ± 0.261.44 ± 0.662.37 ± 0.92<0.001Uric acid, μmol/L228.8 ± 56.3252.9 ± 61.5272.1 ± 65.90.001HbA1c (%) (n = 188)4.91 ± 0.455.21 ± 0.445.42 ± 0.38<0.001TyG index8.17 ± 0.278.49 ± 0.379.08 ± 0.38<0.001Framingham steatosis index (FSI)12.2 ± 0.413.5 ± 0.915.5 ± 0.8<0.001SIC at T1, μg/L 90.0 ± 16.690.0 ± 17.190.3 ± 17.50.992TSH, mIU/L1.28 ± 1.161.31 ± 1.381.37 ± 0.920.060FT3, pmol/L5.01 ± 1.075.14 ± 0.985.36 ± 0.760.011FT4, pmol/L17.98 ± 3.2616.87 ± 2.9215.68 ± 2.53<0.001FT3/FT4 ratio0.281 ± 0.0410.308 ± 0.0500.348 ± 0.061<0.001TPOAb+, n/total (%) *9/34 (20.9%)2/11 (15.4%)0/5 (0%)0.495TGAb+, n/total (%) *4/20 (16.7%)0/6 (0%)0/1 (0%)0.512TRAb+, n/total (%) *0/11 (0%)0/5 (0%)0/2 (0%)N.APregnancy outcomes 



Gestational weeks at delivery, wks39.3 ± 1.339.2 ± 1.638.3 ± 1.5<0.001Birth weight, kg 3.11 ± 0.433.13 ± 0.473.12 ± 0.640.869Fetal distress, n (%)46 (7.8%)26 (6.4%)2 (5.6%)0.640Vaginal delivery, n (%)433 (55.2%)262 (47.9%)20 (36.4%)0.001For continuous variables, data are presented as mean ± SD and were tested by ANOVA or the median (interquartile range, *P*_25_–*P*_75_) and were tested by the Kruskal–Wallis test. For categorial variables, data are expressed as n (%) and were tested by the chi-square test. Mothers with a diagnosis of pre-pregnancy diabetes or hepatic or thyroid disorders were excluded, and 1456 women remained for analysis. Gestational metabolic syndromes (GMSs) were determined to have at least 3 risk factors, including early pregnancy BMI ≥  23.0 kg/m^2^, current gestational diabetes (GDM), current gestational hypertension (GH), or TGs ≥ 1.7 mmol/L. GMS data were available for 1387 mothers. * Thyroid antibodies, including antibodies of thyroperoxidase (TPOAb), TSH receptor (TRAb), and anti-thyroglobulin (TGAb), were only measured by an electro-chemiluminescent immunoassay for mothers with suspicious thyroid dysfunction. Antibodies were regarded as positive if greater than 34 IU/L for TPOAb, 1.75 IU/L for TRAb, and 115 IU/L for TGAb. Abbreviations: SIC, serum iodine concentration; GMS, gestational metabolic syndrome; BMI, body mass index; T1, the first trimester (early pregnancy); GWG, gestational weight gain; TSH, thyroid-stimulating hormone; PCOS, polycystic ovary syndrome; GDM, gestational diabetes mellitus; GH, gestational hypertension; FT3, free triiodothyronine; FT4, free thyroxine; HbA1c, glycosylated hemoglobin A1C; N.A, not applicable. TyG index = In[TGs (mg/dL) × fasting glucose (mg/dL)/2]; FSI, the Framingham steatosis index; FSI = −7.981 + 0.011 × age (years) − 0.146 + 0.173 × BMI at T1 (kg/m^2^) + 0.007 × TGs (mg/dL) + 0.593 × GH (yes = 1, no = 0) + 0.789 × GDM (yes = 1, no = 0) + 1.1 × [ALT/AST ≥ 1.33 (yes = 1, no = 0)].


Table 2 shows the associations of SICs at T1 with metabolic factors by multivariable linear regression. The results of model 1 indicated that LgSIC values at T1 were inversely associated with GWG at T1 (*β* = −0.113, *p* < 0.001) and BMI at T1 (*β* = −0.070, *p* = 0.006). However, when further adjustment was carried out for thyroid markers, the associations were attenuated into non-significance, suggesting that the impact of SICs on body mass is mainly regulated by the thyroid gland. LgSIC values at T1 were positively associated with LgTG (*β* = 0.142, *p* < 0.001), the TyG index (*β* = 0.137, *p* < 0.001), and UA (*β* = 0.060, *p* = 0.018) in model 1. The associations remained significant even with further adjustment for thyroid hormones. Although no significant associations were observed between SICs with 2 h PG, TyG-BMI, and the FSI in model l, they became significant in model 2, with a *β* value of 0.062 (*p* = 0.022) for 2 h PG, a *β* value of 0.073 (*p* = 0.003) for TyG-BMI, and a *β* value of 0.080 (*p* = 0.001) for the FSI. These findings implicate the extra-thyroidal impacts of iodine, in that the thyroid parameters at least partly mediated the associations of SICs with the metabolic markers. The multivariable RCS analyses suggested possible non-linearity (*p* for interaction < 0.1) between SICs at T1 with 1 h PG (*p* = 0.061), 2 h PG (*p* = 0.004), and UA (*p* = 0.089), with an inflection point at 88.9 μg/L of SICs at T1 (Table 1 and Figure 2). However, further stratification analysis showed non-significant findings (Appendix A).
Figure 2The multivariable restricted cubic spline (RCS) between serum iodine concentrations (SICs) and non-linearly related metabolic factors (1 h and 2 h post-load glucose levels and uric acid, with all *p* for nonlinearity < 0.1). Legends: The adjusted variables in the multivariable RCS model were maternal age (y), pre-pregnancy BMI (kg/m^2^), education (primary school and below, middle school, college, university and above), parity (0 ≥ 1), active or passive smoking (yes/no), alcohol drinking (yes/no), medical history of gestational diabetes (yes/no), medical history of gestational hypertension (yes/no), pre-pregnancy PCOS (yes/no), family history of diabetes in first-degree relatives (yes/no), hormone usage (yes/no), total sitting time (h/d), LgTSH, and LgFT3/FT4. Abbreviations: RCS, restricted cubic spline; SIC, serum iodine concentration; T1, the first trimester; BMI, body mass index; PCOS, polycystic ovary syndrome; TSH, thyroid-stimulating hormone.
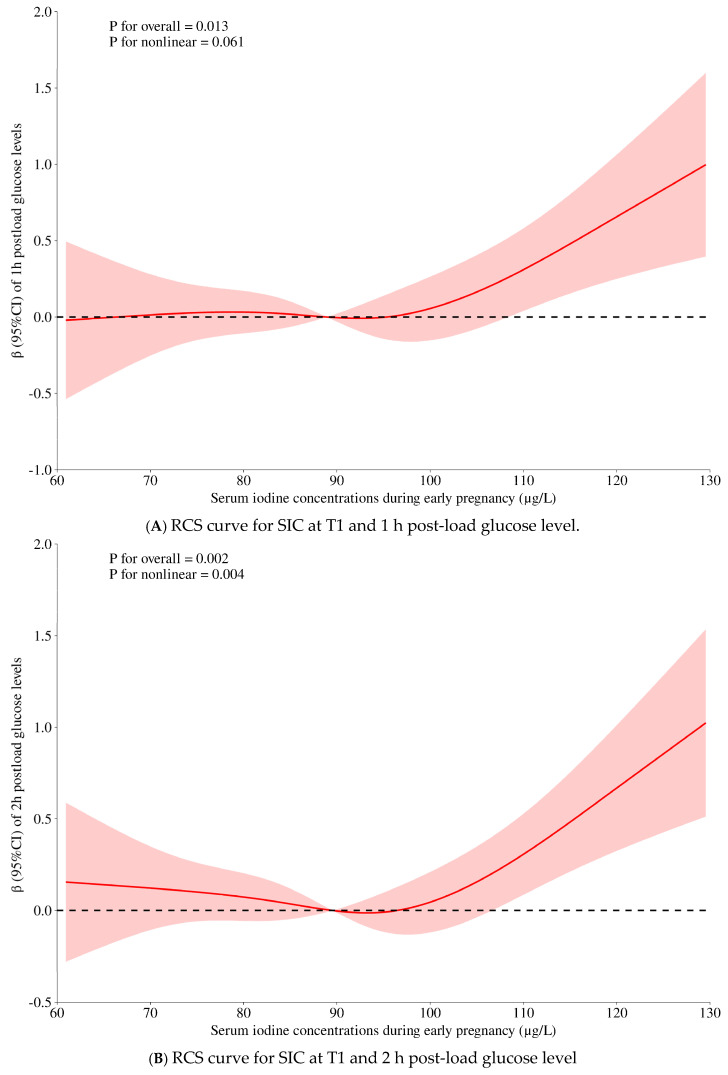

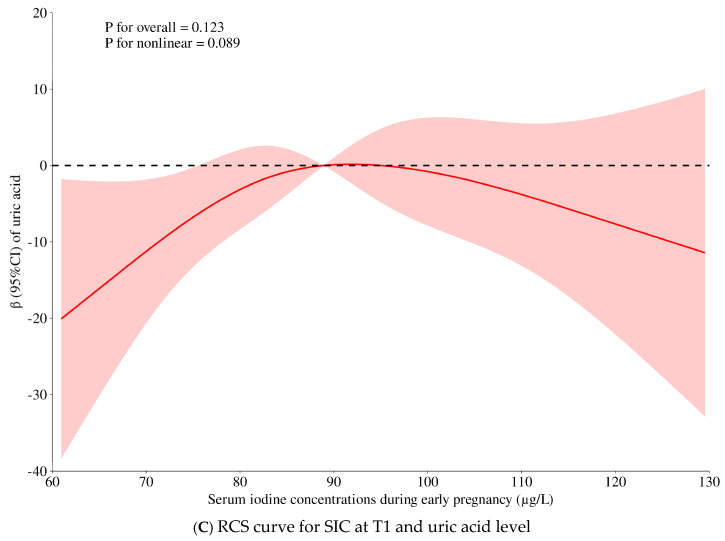

nutrients-17-01626-t002_Table 2Table 2Multivariable linear regression on the associations of serum iodine concentrations (SICs) during early pregnancy (T1) with metabolic factors, Huizhou mother–infant cohort (n = 1456).
Crude ModelModel 1

Model 2 (+Thyroid Markers)*p* forNon-Linearity*B* (95% *CI*)*β**p**B* (95% *CI*)*β**p**B* (95% *CI*)*β**p*GWG at T1 ^#^−3.778 (−5.540, −2.016)−0.110<0.001−3.901 (−5.669, −2.133)−0.113<0.001−0.985 (−2.825, 0.855)−0.0290.2940.231BMI at T1 ^#^−3.247 (−5.712, −0.783)−0.0680.010−3.358 (−5.730, −0.987)−0.0700.0061.849 (−0.463, 4.162)0.0390.1170.683Lg BF% *0.002 (−0.045, 0.049)0.0020.9440.021 (−0.020, 0.062)0.0230.3130.040 (−0.003, 0.084)0.0440.0680.959FBG, mmol/L0.022 (−0.206, 0.251)0.0050.8480.104 (−0.118, 0.326)0.0230.3580.143 (−0.094, 0.381)0.0320.2370.1011-h PG, mmol/L0.105 (−0.920, 1.130)0.0050.8400.459 (−0.535, 1.454)0.0230.3650.880 (−0.180, 1.940)0.0440.1040.0612-h PG, mmol/L0.381 (−0.494, 1.257)0.0220.3930.698 (−0.150, 1.546)0.0410.1071.053 (0.150, 1.956)0.0620.0220.004HbA1c, % *0.072 (−0.031, 0.175)0.0360.1700.082 (−0.022, 0.185)0.0410.1210.095 (−0.016, 0.206)0.0470.0920.108LgTG, mmol/L0.261 (0.157, 0.365)0.128<0.0010.289 (0.193, 0.384)0.142<0.0010.375 (0.275, 0.475)0.185<0.0010.592TC, mmol/L0.301 (−0.133, 0.735)0.0360.1740.334 (−0.100, 0.768)0.0400.1320.406 (−0.057, 0.870)0.0480.0850.420TyG index0.528 (0.298, 0.758)0.117<0.0010.617 (0.406, 0.828)0.137<0.0010.791 (0.569, 1.013)0.176<0.0010.721TyG-BMI ^#^−12.63 (−34.37, 9.11)−0.0300.255−11.55 (−32.47, 9.37)−0.0270.27930.78 (10.27, 51.29)0.0730.0030.668Uric acid, μmol/L41.72 (3.08, 80.37)0.0550.03445.18 (7.86, 82.51)0.0600.01833.95 (−5.98, 73.87)0.0450.0960.089FSI ^#^−0.172 (−0.860, 0.516)−0.0130.624−0.264 (−0.900, 0.372)−0.0210.4161.014 (0.860, 1.163)0.0800.0010.885Mothers with a diagnosis of pre-pregnancy diabetes or hepatic or thyroid disorders or with current hepatic diseases were excluded from the analysis, and 1456 women were analyzed. Serum iodine concentrations (SICs), body fat %, and triglyceride (TG) levels were Log_10_-transformed. Data were analyzed by multivariable linear regression, with covariates being adjusted by the enter method. The adjusted covariates in model 1 included maternal age (y), pre-pregnancy BMI (kg/m^2^), education (primary school and below, middle school, college, university and above), parity (0, ≥1), active or passive smoking (yes/no), alcohol drinking (yes/no), medical history of GDM (yes/no), medical history of GH (yes/no), pre-pregnancy PCOS (yes/no), family history of diabetes in first-degree relatives (yes/no), hormones usage (yes/no), and total sitting time (h/d); in model 2, further adjustment was made for thyroid hormones, including LgTSH and LgFT3/FT4. Gestational metabolic syndrome (GMS) was determined with at least 3 factors among the following 4 items, including early pregnancy BMI ≥ 23.0 kg/m^2^, current gestational diabetes (GDM), current gestational hypertension (GH), or TGs ≥ 1.7 mmol/L. * Data on body fat % and HbA1C were only available for 709 and 188 pregnant women before 20 gestational weeks, respectively. ^#^ For variables of BMI at T1, GWG at T1, the FSI, and TyG-BMI, pre-pregnancy BMI was not adjusted as a covariate due to highly possible collinearity. For the FSI, the variables in the formula were not adjusted in the linear regression models. TyG index = In[TGs (mg/dL) × FPG (mg/dL)/2]; TyG-BMI = TyG index × BMI at T1; FSI = −7.981 + 0.011 × age (years) − 0.146 + 0.173 × BMI at T1 (kg/m^2^) + 0.007 × TGs (mg/dL) + 0.593 × GH (yes = 1, no = 0) + 0.789 × GDM (yes = 1, no = 0) + 1.1 × [ALT/AST ≥ 1.33 (yes = 1, no = 0)]; *p* values for non-linearity were based on the multivariable restricted cubic spline (RCS) with the same covariates as the linear regression model 2. Abbreviations: B, unstandardized coefficient; β, standardized coefficient; Lg, logarithm with the base of 10; BF%, body fat percentage; SIC, serum iodine concentration; BMI, body mass index; T1, the first trimester (early pregnancy); GWG, gestational weight gain; FBG, fasting blood glucose; PG, post-load glucose; HbA1C, glycosylated hemoglobin A1C; TC, total cholesterol; TG, triglycerides; TSH, thyroid-stimulating hormone; PCOS, polycystic ovary syndrome; GDM, gestational diabetes mellitus; GH, gestational hypertension; FSI, the Framingham steatosis index; FT3/FT4, the ratio of free triiodothyronine to free thyroxine.


Multivariate ANOVA (MANOVA) suggested that SICs at T1 were significantly associated with overall metabolic profiles, according to Wilks’ lambda test (*p* < 0.001), with major significance being observed for BMI at T1 (*p* = 0.003), TGs (*p* = 0.023), TC (*p* = 0.041), the TyG index (*p* = 0.001), and TyG-BMI (*p* = 0.060). Further multivariable GLM analyses showed that (Table 3), compared with the low-SIC group (<median), mothers with a high SIC (model 1) had significantly increased body fat levels (*p* < 0.001) and higher levels of HbA1c (*p* = 0.020), TGs (*p* = 0.002), the TyG index (*p* < 0.001), TyG-BMI (*p* < 0.001), and the FSI (*p* = 0.001). Significance was retained even after adjustment for thyroid hormones (all *p* ≤ 0.05). Maternal BMI (*p* = 0.012) and GWG at T1 (*p* < 0.001) were significantly lower and UA levels (*p* = 0.025) were significantly higher in the high-SIC group in model 1. However, they were all attenuated into non-significance after further adjustment for thyroid hormones. GLM analyses comparing SICs (T1) by different metabolic conditions suggested similar findings (Appendix A).
nutrients-17-01626-t003_Table 3Table 3Estimated means, standard errors, and mean differences of metabolic factors by high and low concentrations of serum iodine (< or ≥median 88.9 μg/L) in early pregnancy (T1) by multivariable general linear model (GLM) analysis, Huizhou mother–infant cohort.Metabolic FactorsEstimated Means ± SE (Model 1)*p*Estimated Means ± SE (Model 2)*p*BMI at T1, kg/m^2^
0.012
0.398  Low SIC at T121.8 ± 0.2
21.5 ± 0.2
  High SIC at T121.2 ± 0.2
21.6 ± 0.2
  Mean difference−0.58 ± 0.23
0.19 ± 0.22
GWG at T1, kg
<0.001
0.300  Low SIC at T11.12 ± 0.12
0.94 ± 0.12
  High SIC at T10.52 ± 0.12
0.76 ± 0.12
  Mean difference–0.60 ± 0.17
–0.18 ± 0.17
BF% at T1, %
<0.001
<0.001  Low SIC at T128.7 ± 0.2
28.7 ± 0.2
  High SIC at T130.0 ± 0.2
30.1 ± 0.3
  Mean difference1.32 ± 0.35
1.35 ± 0.36
Fasting glucose, mmol/L0.824
0.846  Low SIC at T14.48 ± 0.02
4.48 ± 0.02
  High SIC at T14.49 ± 0.02
4.49 ± 0.02
  Mean difference0.005 ± 0.024
0.005 ± 0.026
1 h PG, mmol/L0.450
0.191  Low SIC at T17.83 ± 0.08
7.78 ± 0.08
  High SIC at T17.91 ± 0.08
7.93 ± 0.08
  Mean difference0.08 ± 0.11
0.15 ± 0.12
2 h PG, mmol/L0.351
0.200  Low SIC at T16.78 ± 0.07
6.75 ± 0.07
  High SIC at T16.87 ± 0.07
6.88 ± 0.07
  Mean difference0.09 ± 0.10
0.13 ± 0.10
HAb1c at T1, % *
0.020
0.050  Low SIC at T15.13 ± 0.06
5.12 ± 0.06
  High SIC at T15.33 ± 0.06
5.31 ± 0.07
  Mean difference0.203 ± 0.086
0.187 ± 0.094
TGs, mmol/L
0.002
<0.001  Low SIC at T11.17 ± 0.02
1.16 ± 0.02
  High SIC at T11.28 ± 0.02
1.30 ± 0.03
  Mean difference0.109 ± 0.034
0.138 ± 0.036
TC, mmol/L0.094
0.061  Low SIC at T14.12 ± 0.03
4.12 ± 0.03
  High SIC at T14.19 ± 0.03
4.21 ± 0.03
  Mean difference0.074 ± 0.044
0.087 ± 0.047
TyG index
<0.001
<0.001  Low SIC at T18.26 ± 0.02
8.25 ± 0.02
  High SIC at T18.36 ± 0.02
8.37 ± 0.02
  Mean difference0.100 ± 0.024
0.122 ± 0.025
TyG-BMI
<0.001
<0.001  Low SIC at T1177.9 ± 0.4
177.8 ± 0.4
  High SIC at T1180.1 ± 0.4
180.5 ± 0.4
  Mean difference2.19 ± 0.55
2.66 ± 0.58
Uric acid, μmol/L
0.025
0.195  Low SIC at T1238.9 ± 2.7
240.5 ± 2.7
  High SIC at T1247.5 ± 2.7
245.7 ± 2.8
  Mean difference8.63 ± 3.8
5.22 ± 4.03
FSI
0.001
<0.001  Low SIC at T112.7 ± 0.03
12.7 ± 0.03
  High SIC at T112.8 ± 0.03
12.8 ± 0.03
  Mean difference0.12 ± 0.04
0.12 ± 0.04
Data are presented as estimated mean ± SE (standard error). Mothers with a diagnosis of pre-pregnancy diabetes, hepatic or thyroid disorders, or current hepatic diseases were excluded from the analysis, and 1456 women were analyzed. Serum iodine concentrations (SICs) at T1 were dichotomized into low and high groups by a median of 88.9 μg/L. Data were analyzed by a multivariable general linear model (GLM), with covariates being adjusted by the enter method. The mean difference was compared by the LSD approach. The adjusted covariates in model 1 included the following: maternal age (y), pre-pregnancy BMI (kg/m^2^), education (primary school and below, middle school, college, university and above), parity (0, 1, 2, ≥3), smoking (yes/no), alcohol drinking (yes/no), medical history of GDM (yes/no), pre-pregnancy PCOS or thalassemia (yes/no), first-degree family history of diabetes (yes/no), total sitting time (hrs), folate supplementation (yes/no), and hormones usage before and during pregnancy (yes/no); in model 2, further adjustment was made for thyroid hormones, including LgTSH and LgFT3/FT4. In terms of BMI and GWG at recruitment (T1) as dependent variables, pre-pregnancy BMI was not adjusted due to potential collinearity. * Data on body fat % and HbA1C were only available for 709 and 188 pregnant women before 20 gestational weeks, respectively. For the variables BMI at T1, GWG at T1, the FSI, and TyG-BMI, pre-pregnancy BMI was not adjusted as a covariate due to the high possibility of collinearity. For the FSI, the variables in the formula were not adjusted in the linear regression models. TyG index = In[TGs (mg/dL) × FPG (mg/dL)/2]; TyG-BMI = TyG index × BMI at T1; FSI = −7.981 + 0.011 × age (years) − 0.146 + 0.173 × BMI at T1 (kg/m^2^) + 0.007 × TGs (mg/dL) + 0.593 × GH (yes = 1, no = 0) + 0.789 × GDM (yes = 1, no = 0) +1.1 × [ALT/AST ≥ 1.33 (yes = 1, no = 0)]. Abbreviations: BF%, body fat percentage; SIC, serum iodine concentration; PG, post-load glucose level; BMI, body mass index; T1, the first trimester (early pregnancy); GWG, gestational weight gain; TC, total cholesterol; TGs, triglycerides; TSH, thyroid-stimulating hormone; PCOS, polycystic ovary syndrome; GDM, gestational diabetes mellitus; GH, gestational hypertension; FSI, the Framingham steatosis index; FT3/FT4, the ratio of free triiodothyronine to free thyroxine.


### 3.1. The Joint Effects of Maternal SICs (T1) and Metabolic Conditions on Pregnancy Outcomes (Table 4 and Table 5)

We only report the results for a *p* of interaction (the product term of high/low SICs at T1 with metabolic conditions) less than or near 0.1. The multivariable GLM results (Table 4) showed that mothers with a high risk of GMS (≥1 item), pre-pregnancy overweight/obesity (BMI ≥ 23.0 kg/m^2^), or a high FSI (≥median) had significantly increased postpartum bleeding in both the high- and low-SIC groups (all *p* < 0.05). After full adjustment, mothers with both high SICs and high TGs (≥1.7 mmol/L) or hyperlipidemia (TGs ≥ 1.7 or TC ≥ 5.18 mmol/L) had slightly, although significantly, shortened gestational weeks at delivery (both with a mean difference of −0.7 ± 0.2 GW in model 2, both with *p* ≤ 0.001). Mothers with both high SICs at T1 and GH had significantly reduced birth lengths (mean difference of −1.454 ± 0.517 cm in model 2, *p* = 0.005). The results of the multivariable logistic regression (Table 5) showed that, compared with the low-SIC and low-FSI group, mothers with both high SICs and a high FSI at T1 had a reduced risk of SGA (OR = 0.535, 95% CI: 0.322–0.889 in model 2).
nutrients-17-01626-t004_Table 4Table 4The joint effects of maternal serum iodine concentrations (SICs) at T1 and metabolic conditions on various pregnancy outcomes by multivariable general linear models (GLMs), Huizhou mother–infant cohort.
nCrude ModelModel 1Model 2*p* for InteractionPostpartum bleeding, mL



0.104  Low SIC*p*0.0010.0010.011
    No GMS (0 item)288198.3 ± 6.2201.7 ± 4.9203.0 ± 5.1
    High risk of GMS (≥1 item)228229.5 ± 6.9 **225.5 ± 5.5 **223.4 ± 5.7 *
    Mean difference
31.2 ± 9.3 **23.8 ± 7.4 **20.3 ± 8.0 *
  High SIC*p*0.0020.0160.026
    No GMS (0 item)301189.8 ± 5.8193.9 ± 4.8194.1 ± 5.0
    High risk of GMS (≥1 item)216217.3 ± 6.8 **212.3 ± 5.7 *212.4 ± 6.0 *
    Mean difference
27.5 ± 8.918.4 ± 7.6 *18.2 ± 8.1 *
Postpartum bleeding, mL



0.002  Low SIC*p*<0.0010.0010.007
    Pre-pregnancy BMI < 23.0405200.5 ± 5.1204.8 ± 4.1205.1 ± 4.1
    Pre-pregnancy BMI ≥ 23.0131244.4 ± 9.0 **231.7 ± 7.2 **229.2 ± 7.6 **
    Mean difference
43.9 ± 10.326.9 ± 8.3 24.1 ± 8.9, 
  High SIC*p*<0.0010.0010.005
    Pre-pregnancy BMI < 23.0429193.4 ± 4.8195.1 ± 3.9195.5 ± 4.1
    Pre-pregnancy BMI ≥ 23.0108231.0 ± 9.6 **225.2 ± 8.1 **223.2 ± 8.5 **
    Mean difference
37.6 ± 10.7 **30.0 ± 9.2 **27.7 ± 9.7 **
Postpartum bleeding, mL



0.003  Low SIC*p*0.0010.0130.094
    FSI < median264196.6 ± 6.4203.1 ± 5.1205.3 ± 5.4
    FSI ≥ median252228.3 ± 6.6221.7 ± 5.2219.1 ± 5.5
    Mean difference*p*31.6 ± 9.2 **18.6 ± 7.413.8 ± 8.2
  High SIC
<0.001<0.001<0.001
    FSI < median271180.3 ± 6.0186.6 ± 5.1187.4 ± 5.4
    FSI ≥ median 246224.5 ± 6.3 **218.0 ± 5.4 **217.3 ± 5.6 **
    Mean difference
44.3 ± 8.7 **31.4 ± 7.6 **29.9 ± 8.2 **
Delivery weeks, GWs



0.046  Low SIC
0.1300.3360.378
    TGs < 1.7 mmol/L46039.3 ± 0.139.3 ± 0.139.3 ± 0.1
    TGs ≥ 1.7 mmol/L5739.0 ± 0.239.1 ± 0.239.1 ± 0.2
    Mean difference
−0.3 ± 0.2−0.2 ± 0.2−0.2 ± 0.2
  High SIC
<0.001<0.0010.001
    TGs < 1.7 mmol/L45439.3 ± 0.139.3 ± 0.139.3 ± 0.1
    TGs ≥ 1.7 mmol/L6438.4 ± 0.2 **38.6 ± 0.2 **38.5 ± 0.2 **
    Mean difference
−0.9 ± 0.2 **−0.7 ± 0.2 **−0.7 ± 0.2 **
Delivery weeks, GWs



0.043  Low SIC
0.0740.1170.160
    No hyperlipidemia43539.3 ± 0.139.3 ± 0.139.3 ± 0.1
    Yes hyperlipidemia8239.0 ± 0.239.1 ± 0.239.1 ± 0.2
    Mean difference
−0.3 ± 0.2−0.3 ± 0.2−0.2 ± 0.2
  High SIC
<0.001<0.001<0.001
    No hyperlipidemia41939.4 ± 0.139.3 ± 0.139.3 ± 0.1
    Yes hyperlipidemia9938.6 ± 0.238.7 ± 0.2 **38.6 ± 0.2 **
    Mean difference
−0.8 ± 0.2−0.7 ± 0.2−0.7 ± 0.2 **
Birth weight, kg



0.071  Low SIC
0.8260.3140.672
    No hyperlipidemia4353.16 ± 0.023.15 ± 0.023.16 ± 0.02
    Yes hyperlipidemia823.17 ± 0.053.20 ± 0.043.18 ± 0.04
    Mean difference
0.012 ± 0.0550.045 ± 0.0450.019 ± 0.045
  High SIC
0.0150.9470.952
    No hyperlipidemia4193.10 ± 0.023.08 ± 0.023.08 ± 0.02
    Yes hyperlipidemia992.98 ± 0.053.08 ± 0.043.08 ± 0.04
    Mean difference
−0.122 ± 0.0500.003 ± 0.0410.003 ± 0.042
Birth length, cm



0.096  Low SIC
0.6520.4830.871
    No hyperlipidemia43449.9 ± 0.149.8 ± 0.149.9 ± 0.1
    Yes hyperlipidemia8249.8 ± 0.250.0 ± 0.249.9 ± 0.2
    Mean difference
−0.10 ± 0.230.13 ± 0.190.03 ± 0.19
  High SIC
0.0070.8030.777
    No hyperlipidemia41949.8 ± 0.149.7 ± 0.149.6 ± 0.1
    Yes hyperlipidemia9749.1 ± 0.2 **49.7 ± 0.249.7 ± 0.2
    Mean difference
−0.66 ± 0.25 **0.05 ± 0.190.06 ± 0.20
Birth length, cm



0.045  Low SIC
0.4960.0520.189
    TGs < 1.7 mmol/L46049.8 ± 0.149.8 ± 0.149.8 ± 0.1
    TGs ≥ 1.7 mmol/L5750.0 ± 0.250.2 ± 0.250.1 ± 0.2
    Mean difference
0.18 ± 0.260.43 ± 0.220.29 ± 0.22
  High SIC
0.0360.4870.436
    TGs < 1.7 mmol/L45449.7 ± 0.149.6 ± 0.149.6 ± 0.1
    TGs ≥ 1.7 mmol/L6449.1 ± 0.3 *49.8 ± 0.249.8 ± 0.2
    Mean difference
−0.62 ± 0.29 *0.23 ± 0.490.18 ± 0.25
Birth length, cm



0.017  Low SIC
0.8830.0430.104
    Normal51849.8 ± 0.149.8 ± 0.149.8 ± 0.1
    GH1749.8 ± 0.550.6 ± 0.4 *50.4 ± 0.4
    Mean difference
−0.07 ± 0.460.75 ± 0.37 *0.61 ± 0.37
  High SIC
0.0030.0130.005
    Normal52549.7 ± 0.149.7 ± 0.149.7 ± 0.1
    GH1147.7 ± 0.7 **48.5 ± 0.49 *48.2 ± 0.5 **
    Mean difference
−1.97 ± 0.66 **−1.22 ± 0.49 *−1.45 ± 0.52 **
The joint effects of SIC at T1 with metabolic conditions on pregnancy outcomes are reported only for those with a p of interaction ≤ 0.1. Data are presented as estimated mean ± SE and were analyzed by multivariable general linear models (GLMs) with metabolic conditions (yes/no) as the fixed factors. The mean difference was compared by the LSD approach. Hyperlipidemia was defined as either TC > 5.18 mmol/L or TGs > 1.7 mmol/L at T1. For postpartum bleeding, the adjusted covariates in model 1 included maternal age (y), education (5 categories), parity (1, ≥1), medical history of PCOS (yes/no), family history of diabetes (yes/no), delivery modes (vaginal, cesareans, and forceps), and birth weight (kg). In model 2, further adjustment was made for thyroid hormones (LgTSH and LgFT3/FT4). For delivery weeks, the adjusted covariates in model 1 included maternal age (y), pre-pregnancy BMI (kg/m^2^), education (5 categories), parity (0, ≥1), medical history of GH (yes/no) or PCOS (yes/no), family history of diabetes (yes/no), delivery modes (vaginal, cesareans, and forceps), birth weight (kg), and PROM (yes/no). In model 2, further adjustment was made for thyroid hormones (LgTSH and LgFT3/FT4). For birth weight and length, the adjusted covariate in model 1 included maternal age (y), pre-pregnancy BMI (kg/m^2^), education (5 categories), parity (0, ≥1), medical history of GH (yes/no) or PCOS (yes/no), family history of diabetes (yes/no), delivery week (wks), and infant gender (female/male). In model 2, further adjustment was made for thyroid hormones (LgTSH and LgFT3/FT4). **, *p* < 0.01, * *p* < 0.05. Abbreviations: T1, first trimester; GMS, gestational metabolic syndrome; SIC, serum iodine concentration; TGs, triglycerides; GH, gestational hypertension; FSI: the Framingham steatosis index. FSI = −7.981 + 0.011 × age (years) − 0.146 + 0.173 × BMI at T1 (kg/m^2^) + 0.007 × TGs (mg/dL) + 0.593 × GH (yes = 1, no = 0) +0.789 × GDM (yes = 1, no = 0) + 1.1 × [ALT/AST ≥ 1.33 (yes = 1, no = 0)].
nutrients-17-01626-t005_Table 5Table 5The joint effects of serum iodine concentrations (SICs) at T1 and metabolic conditions on the risk of adverse birth outcomes by multivariable logistic regression models, Huizhou mother–infant cohort.
Cases/nCrude Or (95% CI)Model 1 OR (95% CI)Model 2 OR (95% CI)*p* for InteractionSGA



0.107  Low SIC, low FSI58/256111
  Low SIC, high FSI53/2370.983 (0.644, 1.501)0.890 (0.544, 1.455)1.128 (0.653, 1.948)
  High SIC, low FSI89/266111
  High SIC, high FSI55/2340.611 (0.412, 0.907)0.580 (0.360, 0.934)0.535 (0.322, 0.889)
LGA



0.095  Low SIC, normal lipids16/340111
  Low SIC, hyperlipidemia7/662.373 (0.936, 6.018)1.905 (0.681, 5.332)1.691 (0.596, 4.801)
  High SIC, normal lipids15/300111
  High SIC, hyperlipidemia2/730.578 (0.128, 2.601)0.464 (0.099, 2.182)0.424 (0.089, 2.029)
LBW



0.098  Low SIC, normal BP27/518111
  Low SIC, GH3/174.157 (1.122, 15.409)3.011 (0.383, 23.709)3.372 (0.413, 27.534)
  High SIC, normal BP34/525111
  High SIC, GH2/113.591 (0.733, 17.594)1.929 (0.084, 44.125)1.893 (0.080, 44.778)
The analysis was carried out using multivariable logistic regression. For SGA by the FSI, the adjusted covariates included maternal age (y), education (5 categories), parity (0, ≥1), medical history of PCOS (yes/no), and family history of diabetes (yes/no). For model 2, further adjustment was made for thyroid hormones (LgTSH and LgFT3/FT4). For LGA by hyperlipidemia and LBW by GH, model 1 was additionally adjusted for pre-pregnancy BMI (kg/m^2^). For LBW, the adjusted covariates included maternal age (y), pre-pregnancy BMI (kg/m^2^), education (5 categories), parity (0, ≥1), medical history of PCOS (yes/no), family history of diabetes (yes/no), and delivery weeks (wks). For model 2, further adjustment was made for thyroid hormones (LgTSH and LgFT3/FT4). Abbreviations: SIC, serum iodine concentration; FSI, the Framingham steatosis index; SGA, small for gestational age; LGA, large for gestational age; LBW, low birth weight.


### 3.2. Sensitivity and Subgroup Analyses

The sensitivity analyses among the women with euthyroid (n = 1307), non-GH (n = 1436), normal pre-pregnancy BMI (<23 kg/m^2^, n = 1337), and normo-lipidemia (n = 1152) suggested similar findings to those of all participants in terms of the associations of SICs at T1 with metabolic factors (Appendix A). Subgroup analyses by GDM (yes vs. no), the TyG index (< vs. ≥8.73), and parity (0 vs. ≥1) suggested more evident associations in mothers with GDM, multi-parity, and insulin resistance (Appendix A).

## 4. Discussion

### 4.1. Current Findings and Implications

Our findings, obtained in an iodine-replete region of China, suggest that mothers with high SICs during early pregnancy were associated with adverse metabolic profiles and increased metabolic disorders. Iodine affects metabolic components through both thyroid and extra-thyroid pathways. Mothers with metabolic conditions had an increased risk of postpartum hemorrhage under either a high or low iodine status. Mothers with both high SICs and GH were associated with a notably reduced birth length. Both high SICs and hyperlipidemia might result in a shortened gestational duration, although high SICs and a high FSI might have a reduced risk of SGA. Our findings confirm that simultaneously monitoring maternal iodine status and metabolic components during early pregnancy could help in the early identification of obstetric complications and adverse birth outcomes. To our knowledge, this study was the first to utilize serum iodine as an individual iodine biomarker to test the extra-thyroidal impacts of iodine on metabolic factors during gestation and to examine their joint effects on common pregnancy outcomes. These findings, obtained in an iodine-sufficient region, stress that pregnant women with both high SICs and metabolic abnormalities should receive special attention in regular prenatal visits. Iodine excess and the extra-thyroidal effects of iodine have received increasing attention, especially in iodine-replete regions and vulnerable subgroups with a universal iodine fortification of more than 30 years. Our findings emphasize that iodine prophylaxis programs (such as salt iodization and routine iodine prescription during pregnancy) in iodine-sufficient regions require careful surveillance among mothers with comorbid metabolic conditions.

### 4.2. Maternal Iodine with Metabolic Components

Our findings were consistent with reports in both Asian [8,23] and Western adults [24,25] showing that a high iodine load might increase the risk of metabolic abnormalities. In contrast, studies in iodine insufficient or moderate regions have reported inconsistent findings with ours, showing that high UICs, dietary iodine intake, or iodine supplementation were associated with favorable metabolic profiles, including lowered lipids levels, decreased glucose or insulin resistance, reduced risk of obesity, or lowered uric acid levels [26,27,28,29]. A U-shaped association was suggested between iodine status and metabolic disorders in both the general population [23] and pregnant mothers [30]. Evidence indicates that iodine contributes to redox balance functioning as either a prooxidant (under an insufficient or excessive iodine status) or antioxidant (under appropriate iodine levels) [8,23,28,30]. A large-scale cross-sectional study (Thyroid disease, Iodine nutrition and Diabetes Epidemiology, TIDE) among 74,100 Chinese adults [23] reported that both low and high UICs increased the risk of metabolic conditions, with the lowest risk being at 300–499 μg/L. Another study among euthyroid pregnant Chinese women in an iodine-sufficient area also revealed that [30] both mild iodine deficiency and excess during early pregnancy had adverse pregnancy outcomes (GDM and macrosomia). Several studies utilizing NAHANSE data have reported controversial findings in the US population. Some suggested that a low UIC was associated with increased metabolic disorders, including dyslipidemia [29], hyperglycemia [31], and cardiometabolic abnormalities [32], while others did not [28,33]. The discrepancy in the findings based on NHANSE data might be due to its cross-sectional design, the application of UICs for personal iodine assessment, the high prevalence of iodine deficiency, and the overall declining trend of populational iodine status.

The physio-pathological mechanisms of iodine in metabolic disorders are still unclear. Although thyroid dysfunction was suggested to be an important reason for metabolic abnormity [34], our findings provide additional evidence that high gestational iodine levels, independent of thyroid hormones, were associated with metabolic abnormalities even in euthyroid mothers. The impacts of iodine on metabolic factors could be through both thyroidal and non-thyroidal pathways [3]. An appropriate physiological iodine level can balance oxidative homeostasis, whereas excessive iodine exposure might aggregate oxidative stress and chronic inflammation, predisposing patients to increased metabolic disorders [6]. The joint analyses in our study further indicate that elevated SICs and the presence of metabolic conditions simultaneously (i.e., hyperlipidemia and GH) might have resulted in the increased risk of adverse pregnancy outcomes. Increased inflammation conferred from mothers with metabolic aberrations to the fetus or dysbiosis of the gut microbiota might explain the increased risk of adverse pregnancy outcomes [35].

### 4.3. Maternal Iodine, BMI, and the Risk of Postpartum Hemorrhage

The present findings confirmed our previous data obtained from 1101 euthyroid mothers, showing that a high iodine load was associated with lowered maternal weight gain [13]. The impacts of iodine on maternal body mass were mainly regulated by thyroid hormones. The findings were also in line with the TIDE study [23] and the NHANSE results [28], showing that iodine status was inversely associated with the risk of obesity. Iodine is an essential nutrient for regulating growth, development, and metabolism via the biosynthesis of thyroid hormones, and it exerts direct effects on energy metabolism. When iodine is sufficient, adequate thyroid hormones may promote the body’s energy expenditure or decrease its energy intake. Further joint analyses suggested that low maternal SICs combined with GMS or a high pre-pregnancy BMI resulted in significantly increased postpartum bleeding, even after controlling for birth weight, delivery modes, and thyroid hormones. This might have been due to low maternal iodine levels interacting with metabolic conditions, resulting in excessive GWG and an increased birth size, which can cause increased bleeding at delivery [28,36].

### 4.4. Maternal Iodine, Glycemic Control, and Pregnancy Outcomes

In contrast with the inverse association of maternal SICs with body mass, SICs in early pregnancy were positively associated with both 2 h PG and the TyG index, even after adjusting for thyroid markers, with a more evident relationship being observed in euthyroid mothers. Similar to our findings, a cross-sectional study in Shanxi province, China, also reported that a long-term intake of excess iodine might lead to metabolic disorders, including elevated glucose, BP, and lipids [8]. Our findings suggest that high iodine levels in early gestation may potentially increase the risk of glucose intolerance and insulin resistance through both thyroidal and non-thyroidal pathways. Thyroid hormones, which require iodine for their biosynthesis, play an important role in regulating the process of glucose disposal [37]. Triiodothyronine (T3) not only directly stimulated the insulin-mediated uptake of basal glucose but also increased insulin-stimulated glucose transport by inducing an increase in GLUT4 protein levels in the skeletal muscles of euthyroid rats [37]. In addition, a high iodine load could directly precipitate oxidative stress and arouse an imbalanced redox reaction, thereby aggregating hyperglycemia [6,7]. Although a high maternal SIC was associated with glucose intolerance, we did not observe a significant interaction between SICs and GDM on adverse pregnancy outcomes. This might have been due to the fact that when detected, GDM undergoes tight management by either dietary modification or medication treatment, which could restrict maternal weight gain and normalize fetal growth, providing similar neonatal outcomes as nondiabetic pregnancies [38].

### 4.5. Maternal Iodine, Lipids, and Gestational Duration

Our results indicated that maternal SICs were positively associated with lipid levels. These findings are consistent with several observational reports in general populations [23,28] but not all [39]. The TIDE study observed a positive association between the UIC and dyslipidemia when the UIC was above 300 μg/L [23]. Another two observational reports indicated that high water iodine levels might increase the risk of dyslipidemia, such as elevated LDL-c, TGs, and TC or reduced HDL-c [8,40]. However, a randomized controlled trial in overweight women reported a reduced incidence of hypercholesterolemia with iodine supplementation [39]. Early pregnancy is a key stage of placenta formation and embryonic organogenesis, and any disturbances in lipid metabolism during this vulnerable period might lead to adverse birth outcomes [41]. Thyroid hormones and TSH play significant roles in lipid metabolism by enhancing lipid synthesis or lowering LDL-c clearance [42]. An animal experiment provided evidence for the fact that long-term high water iodine intake could result in dose- and time-dependent hypercholesterolemia, resulting from TRbeta1-mediated downregulation of the hepatic LDLr gene induced by excess iodine ingestion [43]. We further observed that both high SICs and hyperlipidemia were associated with a significantly reduced gestational duration (0.7–0.9 week) at delivery. The magnitude of the shortened duration (5–6 days) was comparable to a meta-analysis [44] (21 randomized controlled trials and 10,802 pregnant women) on dietary fish oil supplementation, which reported a 5.8-day increase in gestational age, a 22% reduced risk for early preterm delivery, a 23% lowered risk of LBW, and an increased infantile birth weight by 51.2 g. Whether the shortened gestation duration with high SICs and high lipids in our data implicated an increased risk of preterm delivery or LBW is unknown. These findings need to be confirmed in a large-scale birth cohort. Evidence suggests that gestational dyslipidemia was associated with an increased risk of preterm birth [41]. Hyperlipidemia not only relates to endothelial oxidative stress and inflammation but also raises non-esterified fatty acid availability for fetal growth via placental transfer and increases the risk of LGA, even in non-obese mothers [45]. In contrast, our findings showed that a high SIC was associated with decreased GWG and an increased risk of SGA [46], which could help explain the fact that both high SICs and high lipids levels in our study did not significantly affect birth weight.

### 4.6. Maternal Iodine, the FSI, and the Risk of SGA

We observed that high maternal SICs were associated with a higher FSI, possibly increasing the risk of hepatic steatosis. This association remained significant in healthy mothers (devoid of hepatic disorders or thyroid dysfunction). The FSI is a comprehensive index that not only comprises variables for metabolic conditions (obesity, hypertension, and diabetes) but also markers of liver function (AST/ALT). Our study was the first to observe that a high maternal SIC might increase the risk of gestational hepatic steatosis. A recent report based on NHANES data also suggested a positive association between the UIC and NAFLD risk in general populations [33]. The liver plays a key role in lipid and glucose metabolism, but excessive iodine was reported to disrupt the structure, metabolism, and function of cell membranes and cause liver damage [47,48]. Excessive iodine could induce oxidative stress and disturb thyroid hormone metabolism by upregulating mRNA expression for SREBP-1c and FAS (fatty acid synthase) in the liver, which plays a pivotal role in hepatic steatosis [49]. Our results further showed that mothers with both high SICs and a high FSI during early pregnancy had a reduced risk of SGA, with the associations being independent of thyroid hormones. The possible reason might be as follows: First, hepatic steatosis is closely correlated with insulin resistance and hepatic triglyceride accumulation [50], which might increase the accumulation of lipids in the fetus and reduce the risk of SGA. Second, the metabolic conditions contained in the FSI formula might have synergistic interactions predisposing to increased fat levels in the fetus [1,38]. We did not observe an obvious impact of both a high SCI and high FSI on LGA risk. This might have been due to the fact that we had limited LGA cases at delivery. In addition, most of the mothers had a relatively normal range of BMI and the FSI. The association between maternal SICs, fatty liver, and birth outcomes deserves in-depth investigation in large-scale prospective studies.

### 4.7. High Maternal SICs and GH and Lowered Birth Length

We did not observe a significant association between SICs and GH risk. However, the combination of high SICs and GH incidence showed a significantly reduced birth length, with a mean difference of −1.454 ± 0.517 cm. GH, or chronic hypertension, has been reported to cause SGA while reducing LGA risk [38,51]. The associations were independent of concurrent glucose disturbances. GH implicates circulatory maladjustments during pregnancy, often being paralleled with suboptimal placentation and increased endothelial shear. All these factors might contribute to attenuated fetal growth or an increased risk of preterm birth [38]. High iodine exposure puts cells in a state of high oxidative stress, which appears to synergize with GH and aggregate chronic inflammation states (the basic pathophysiology of metabolic conditions), subsequently affecting fetal growth and development.

### 4.8. Study Strengths and Limitations

The major strengths of this study were the prospective design, the employment of SICs for the evaluation of individual iodine status, and the exploration of the joint impacts of maternal iodine and metabolic components on pregnancy outcomes. We excluded the influence of thyroid dysfunction and adjusted for thyroid hormones in all the regression models, and we determined an independent extra-thyroidal effect of iodine on metabolic factors and birth outcomes.

Several limitations of our study merit discussion. First, serum iodine and metabolic markers were tested in early pregnancy. These cross-sectional data have limitations in terms of causality inference. Further studies monitoring the dynamic changes in these biomarkers over gestation are necessary to confirm the interplay of SICs and metabolic factors. Second, the optimal ranges of maternal SICs in the prevention of metabolic disorders and adverse birth outcomes could not be confirmed from the current analysis. This was due to the generally linear relationship between iodine and metabolic variables and the mostly adequate, even excessive, iodine status in our participants. We thus only observed the right side of the U-shaped curve. Third, due to resource limitations, several metabolic components of GMS were unavailable in our data, including waist circumferences and lipid profiles other than TGs and TC. Insulin levels and cytokines related to oxidative stress or inflammation were not tested either. In addition, data on dietary iodine-rich food intakes and the usage of iodized salt were only available in a small group of participants. Future exploration over a wide coverage of involved biomarkers and investigation of detailed dietary iodine sources could substantiate the current findings and exhibit comprehensive perspectives. Finally, our cohort was based on one hospital dataset and only consisted of pregnant Chinese women, which affects the study’s generalizability. Though we controlled for many important covariates, uncontrolled or unmeasured residual factors were unavoidable. Future well-designed large-scale birth cohorts with the inclusion of multiple centers are imperative to corroborate the present findings.

## 5. Conclusions

Our prospective data in iodine-adequate Chinese pregnant women showed that high maternal SICs during early pregnancy were associated with an increased metabolic risk and subsequent adverse birth outcomes. The findings emphasize the importance of individual iodine assessment during gestation in the prevention of adverse birth outcomes, especially among mothers with metabolic abnormalities.

## Figures and Tables

**Figure 1 nutrients-17-01626-f001:**
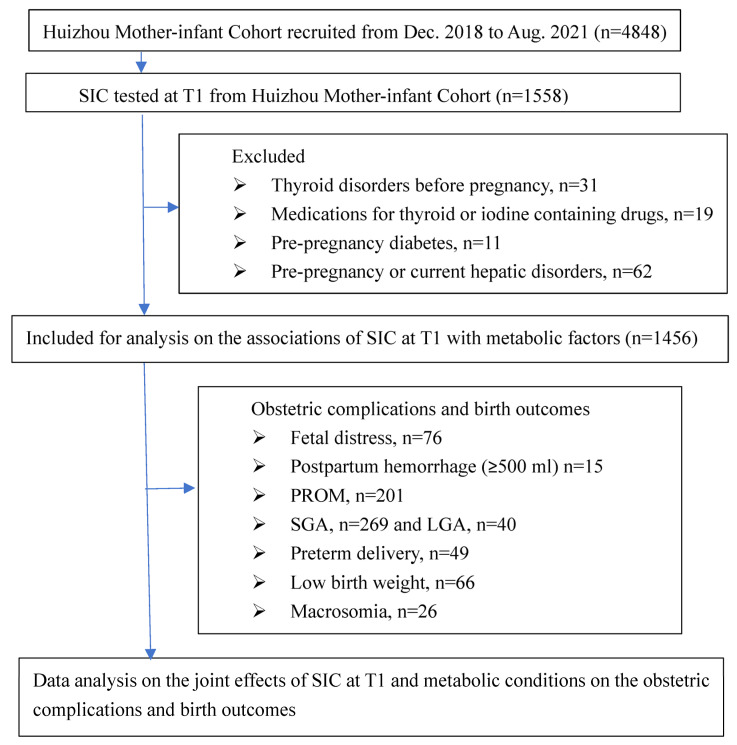
Study flow chart of participant recruitment and data analysis based on Huizhou mother–infant cohort. Legends: SIC, serum iodine concentration; T1, the first trimester; PROM, premature rupture of membrane; SGA and LGA, small and large for gestational age.

## Data Availability

The original datasets are not publicly available due to ethical limitations regarding publishing medical record data, but they are available from the corresponding author on reasonable request under a strict confidential process.

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
