# Peer review of "Extra-Thyroidal Impacts of Serum Iodine Concentrations During Early Pregnancy on Metabolic Profiles and Pregnancy Outcomes: Prospective Study Based on Huizhou Mother–Infant Cohort"

_nutrients, 2025, doi:10.3390/nu17101626_

Round 1
Reviewer 1 Report
Comments and Suggestions for Authors
I would like to congratulate the authors for this fascinating study.
I have a few questions, however.
What was the iodine intake before pregnancy? During pregnancy ?
When were iodine and thyroid hormone levels measured? Please specify gestational weeks. Do you have information on thyroid antibodies?
The TSH levels are higher in women with more components of the metabolic syndrome. How do you know the negative effects of high iodine levels are not due to lower thyroid function?
Please include in Tables ft4 and fT3 levels and not only the ratio, and include Iinitials used must be explained beneath each table and figure.
The English must be improved.
Comments on the Quality of English Language
The quality of the English is irregular, and quite good in some paragraphs, and frankly poor in others. It must be improved.
Author Response
We greatly appreciate all the Reviewers for their valuable comments and feedbacks to improve our work.
Comment 1: I would like to congratulate the authors for this fascinating study.
R: We much appreciate the positive comment from the Reviewer.
Comment 2: I have a few questions, however, What was the iodine intake before pregnancy? During pregnancy?
R: Thanks for the question. This is the limitation of the study. We did not conduct detailed dietary investigation due to limited personnel and time in field data collection. Dietary intake of seaweed and kelp (common iodine rich foods in coastal city) and usage of ionized salt during pre- and early pregnancy were investigated via questionnaire survey, but the answers were available in part of participants. We have added these data in the Table 1 and admitted this limitation in the discussion as (Page 18 line 597-600):
“In addition, data on dietary iodine-rich food intakes and the usage of iodized salt were only available in a small group of participants. Future exploration for a wide coverage of involved biomarkers and investigation of detailed dietary iodine sources could substantiate current findings and exhibit comprehensive perspectives.”
Comment 3: When were iodine and thyroid hormone levels measured? Please specify gestational weeks. Do you have information on thyroid antibodies?
R: Thanks for the question. Serum iodine and thyroid hormones were tested in participants’ first prenatal visit, mostly in the early pregnancy with a median of 12.4 gestational weeks. The time information for biochemical testing has been added in Table 1. Thyroid antibodies including antibodies of thyroperoxidase (TPOAb), TSH receptor (TRAb), and anti-thyroglobulin (TGAb) were measured only for mothers with suspicious thyroid dysfunction. We have also added this information in the Table 1 and its footnotes.
Related sentences have been added in footnotes of Table 1 as: “*Thyroid antibodies including antibodies of thyroperoxidase (TPOAb), TSH receptor (TRAb), and anti-thyroglobulin (TGAb) were measured by electro-chemiluminescent immunoassay only for mothers with suspicious thyroid dysfunction. Antibodies were regarded as positive if greater than 34 IU/L for TPOAb, 1.75IU/L for TRAb, and 115 IU/L for TGAb.”
Comment 4: The TSH levels are higher in women with more components of the metabolic syndrome. How do you know the negative effects of high iodine levels are not due to lower thyroid function?
R: Thanks for the comments. Yes, the TSH levels were a little higher in women with more metabolic components. We thus adjusted thyroid markers (TSH, FT3/FT4) in all the regression models (model 2) to testify the thyroid-independent associations of iodine with metabolic factors and pregnancy outcomes, even though mothers of clinical thyroid dysfunction have been removed before data analysis. Our results showed that, for early GWG and BMI at T1, their significant relationship with SIC were attenuated into non-significance which suggested the impact of SIC on body mass be dominantly regulated by thyroid gland. For TG, UA, TyG-index, 2hPG, TyG-BMI and FSI, the significance was shown after adjustment for thyroid markers, which implicated the extra-thyroidal impacts of iodine that thyroid parameters at least partly mediated the associations of SIC with the metabolic markers.
Comment 5: Please include in Tables ft4 and fT3 levels and not only the ratio, and include Iinitials used must be explained beneath each table and figure.
R: Thanks for the suggestion. The results of baseline FT3 and FT4 by different GMS items have been included in the Table 1, and abbreviations have been denoted in the footnotes. The reason that we adjusted FT3 to FT4 ratio (not the individual FT3 and FT4 levels) was due to FT3 to FT4 ratio reflects not only the deiodinase activity1 (the conversion of FT4 to FT3), but also the peripheral thyroid sensitivity which notably related with metabolic conditions2 and predicts metabolic conditions better3, 4.
Comment 6: The English must be improved.
The quality of the English is irregular, and quite good in some paragraphs, and frankly poor in others. It must be improved.
R: Thanks for the suggestion. We have polished our manuscript carefully and corrected the grammatical, styling and typos found in the manuscript.
References
- Zhang, S.; Wu, Y.; Pan, W.; Li, G.; Zhang, D.; Li, S.; Huang, Q.; Liu, Z. M., Free-Triiodothyronine to Free-Thyroxine Ratio Mediated the Effect of Prepregnancy Body Mass Index or Maternal Weight Gain During Early Pregnancy on Gestational Diabetes Mellitus. Endocr Pract 2022, 28 (4), 398-404.
- Roef, G. L.; Rietzschel, E. R.; Van Daele, C. M.; Taes, Y. E.; De Buyzere, M. L.; Gillebert, T. C.; Kaufman, J. M., Triiodothyronine and free thyroxine levels are differentially associated with metabolic profile and adiposity-related cardiovascular risk markers in euthyroid middle-aged subjects. Thyroid 2014, 24 (2), 223-31.
- Park, S. Y.; Park, S. E.; Jung, S. W.; Jin, H. S.; Park, I. B.; Ahn, S. V.; Lee, S., Free triiodothyronine/free thyroxine ratio rather than thyrotropin is more associated with metabolic parameters in healthy euthyroid adult subjects. Clin Endocrinol (Oxf) 2017, 87 (1), 87-96.
- Jin, Q.; Huang, G.; Tian, X.; Shu, Y.; Tusongtuoheti, X.; Mao, Y., High free triiodothyronine, and free-triiodothyronine-to-free-thyroxine ratio are associated with metabolic syndrome in a euthyroid employee population: the Zhejiang Zhenhai study. Endocr Connect 2023, 12 (5).
Reviewer 2 Report
Comments and Suggestions for Authors
Thank you for the opportunity to review the manuscript titled: Extra-thyroidal Impacts of Serum Iodine Concentrations During Early Pregnancy on Metabolic Profiles and Pregnancy Outcomes- A Prospective Study Based on the Huizhou Mother-Infant Cohort. The manuscript is well-written and includes a good literature review. However, there appears to be some overlap with a previously published study on the same topic. It may be helpful for the authors to clarify how this manuscript builds upon or differs from their earlier publication and to specify any additional metabolic parameters that were analyzed in the current study.
Additionally, in the abstract, the acronym “ICP-MS” needs to be spelled out as “inductively coupled plasma mass spectrometry” upon first use. It would also be beneficial to define the range of high serum iodine concentrations in women that were associated with maternal complications. Finally, the authors may wish to review the tables for consistency in decimal formatting, as this could enhance clarity and a uniform presentation of Tables throughout the manuscript.
Author Response
We greatly appreciate all the Reviewers for their valuable comments and feedbacks to improve our work.
Comment 1: Thank you for the opportunity to review the manuscript titled: Extra-thyroidal Impacts of Serum Iodine Concentrations During Early Pregnancy on Metabolic Profiles and Pregnancy Outcomes- A Prospective Study Based on the Huizhou Mother-Infant Cohort. The manuscript is well-written and includes a good literature review.
R: We appreciate the positive comments from the reviewer.
Comment 2: However, there appears to be some overlap with a previously published study on the same topic. It may be helpful for the authors to clarify how this manuscript builds upon or differs from their earlier publication and to specify any additional metabolic parameters that were analyzed in the current study.
R: Thanks for the comments. As we have elaborated in the introduction: Although several observational studies reported the relationship between iodine and metabolic factors, the conclusions remained speculative with limited studies being conducted among pregnant women. Currently, no study has explored the joint impacts of maternal iodine and metabolic conditions on pregnant outcomes. Most of previous studies applied urinary iodine (a biomarker recommended by WHO for assessment of populational iodine status) for evaluation of individual iodine status. Its high variability with drinking water, consumed foods and circadian rhythm was suspected to be the major reason underlying the controversial findings. Compared with urinary iodine, serum iodine is quite stable and can reflect the bioactive iodine component. However, observational studies utilizing serum iodine as individualized iodine marker were much limited. In addition, most of previous studies on iodine and metabolic factors were conducted in iodine insufficient regions. There are increasing concerns on the adverse health impact from excessive iodine intake as the universal salt iodization policy has implemented more than 30 years. We therefore aimed to examine the associations of gestational SIC, metabolic factors during early pregnancy with subsequent obstetric complications and fetal growth abnormalities in an iodine replete region. The findings help elucidating the extra-thyroid impacts of iodine on gestational metabolic components and subsequent birth outcomes.
Previous studies mostly focused on common metabolic factors, such as obesity, glucose and lipids, while our study additionally included some composite indices such as TyG-index, TyG-BMI and Framingham steatosis index (FSI). As we described in discussion, our study was the first that observed high maternal SIC might increase the risk of gestational hepatic steatosis.
Comment 3: Additionally, in the abstract, the acronym “ICP-MS” needs to be spelled out as “inductively coupled plasma mass spectrometry” upon first use. It would also be beneficial to define the range of high serum iodine concentrations in women that were associated with maternal complications.
R: Thanks for the comments and suggestion. However, we have checked the journal of Nutrients allows using the abbreviation format of ICP-MS in the abstract as it is a common approach for minerals’ testing.
It is a pity that our study currently cannot define the optimal range of SIC for prevention of maternal complications, although we have published the trimester specific reference intervals for SIC on the basis of Huizhou Mother-infant Cohort (Biol Trace Elem Res. 2024 Jun;202(6):2457-2465.). As we have discussed the limitation in page 18 line 590-595 that, “the optimal ranges of maternal SIC in prevention of metabolic disorders and adverse birth outcomes could not be confirmed from current analysis. This was due to the generally linear relationship between iodine and metabolic variables, and the mostly adequate even excessive iodine status in our participants. We thus only observed the right side of the U-shaped curve.”
Comment 4: Finally, the authors may wish to review the tables for consistency in decimal formatting, as this could enhance clarity and a uniform presentation of Tables throughout the manuscript.
R: Thanks for the suggestion. The tables have been checked for the decimals and modified accordingly.
Reviewer 3 Report
Comments and Suggestions for Authors
This manuscript summarizes the associations of gestational serum iodine
concentrations (SIC), metabolic factors during early pregnancy, with
subsequent obstetric complications and fetal growth abnormalities in an iodine-replete region from China. The data can be used to give a new insight and
bring to light new correlations that were first evaluated in the manuscript.
However, only minor revisions are necessary.
- Please change ‘esp.’ with its extensive form everywhere in manuscript.
2. The ‘Introduction’ section is well organized and reflects the aim of the study.
3. Please, provide citations for subchapter ‘2.4. Obstetric and birth outcomes’.
Author Response
We greatly appreciate all the Reviewers for their valuable comments and feedbacks to improve our work.
Comment 1: This manuscript summarizes the associations of gestational serum iodine concentrations (SIC), metabolic factors during early pregnancy, with subsequent obstetric complications and fetal growth abnormalities in an iodine-replete region from China. The data can be used to give a new insight and bring to light new correlations that were first evaluated in the manuscript.
R: We greatly appreciate the positive comments from the Reviewer.
Comment 2: However, only minor revisions are necessary.
- Please change ‘esp.’ with its extensive form everywhere in manuscript.
R: Thanks for the suggestion. All the ‘esp.’ in the main text have been changed as ‘especially’ accordingly.
- The ‘Introduction’ section is well organized and reflects the aim of the study.
R: Many thanks for the positive comment from the Reviewer.
- Please, provide citations for subchapter ‘2.4. Obstetric and birth outcomes’.
R: Thanks for the suggestion. All the diagnostic criteria for obstetric complications were referred to the Chinese textbook of Obstetrics and Gynecology. The statement has been added in the section 2.4 (page 5 line 182-184) as:
“All the diagnostic criteria for obstetric complications were in accordance with the Chinese textbook of Obstetrics and Gynecology (10th edition, The People's Health Press Co., Ltd).”
Reviewer 4 Report
Comments and Suggestions for Authors
Introduction
I.
I recommend a more structured flow for the Introduction:
- Start with the importance of gestational metabolic syndrome (GMS) in relation to pregnancy and metabolic disorders.
- Then transition to the role of iodine, emphasizing the increased requirements during pregnancy and the limitations of current recommendations.
- Highlight the extrathyroidal functions of iodine, which open new research directions.
- Point out the knowledge gap: limited evidence regarding serum iodine concentration (SIC), especially among pregnant women in iodine-adequate areas.
- Conclude with a clear statement of the study objective.
II.
Line 56: “Recent studies suggested that iodine, aside from its role in the thyroid, functions as an antioxidant, immunomodulator, and differentiator in various organs and tissues.”
It would be beneficial to include concrete examples for these functions.
Materials and Methods
III.
The FSI and TyG indices should be briefly defined upon first mention to improve clarity.
Also, the formula formatting should be improved for readability, e.g., TyG-index = ln[TG (mg/dL) × FPG (mg/dL) / 2].
IV.
The use of multiple statistical software packages is appropriate. However:
- A) A more detailed description of the data transformation process would be helpful—how was normality assessed, and were any other transformations applied?
- B) It should be indicated which software was used for which analysis (e.g., R for restricted cubic spline, SPSS for regression models).
- C) The specific statistical tests used to assess group differences should be explicitly stated (e.g., ANOVA, Kruskal–Wallis, chi-square test).
- D) Regarding the restricted cubic spline (RCS): the rationale behind choosing the 0–35–65–95 percentiles should be clarified. If the distribution was skewed or included outliers, this could influence the spline modeling. Furthermore, it would be standard to report effect sizes and confidence intervals, not only p-values < 0.1.
Results
V.
In Table 1, it is not specified which statistical tests were used to evaluate group differences. This information should be clearly stated (e.g., t-test, Mann–Whitney U, chi-square test, or ANOVA).
VI.
Important findings are often referred to as “data not shown,” and numerical results (e.g., β coefficients, confidence intervals, p-values) are rarely presented, especially in mediation and sensitivity analyses. Some interpretations may overstate the strength of associations derived from cross-sectional data.
Discussion
VII.
Section 4.2 (“Maternal iodine with metabolic components”) is overly long and includes repetitive elements. Some statements closely repeat content from the Introduction. A more concise presentation is recommended to maintain focus.
VIII.
In section 4.3 (“Maternal iodine, BMI and postpartum hemorrhage”), the proposed mechanism—excessive gestational weight gain (GWG) → larger fetus → postpartum hemorrhage—is speculative. Notably, no significant association was found with birth weight in the study.
IX.
Section 4.5 (“Maternal iodine, lipids and gestational duration”):
- A) The reported decrease in gestational duration (0.7–0.9 weeks) is considered significant, yet its clinical relevance is questionable and not fully discussed.
- B) The oxidative stress hypothesis is reintroduced without empirical support, as no oxidative markers were measured in the study.
Conclusions
X.
The Conclusions section is concise, but some expressions may be overstated (e.g., “clues to pathogenesis”). It does not discuss potential clinical applicability or implications for future guidelines. Although defining optimal SIC ranges appears to be an implicit aim of the study, this is not addressed in the conclusions.
Minor Issues
XI.
In Figure 1, the labels “Low birth weight, n = 66” and “macrosomia, n = 26” are not clearly visible.
XII.
Lines 44–46: The sentence “Identification of novel nutritional factors therefore received numerous attentions for prevention and control of metabolic disorders during pregnancy.” should be revised to:
“Therefore, nutritional factors have received considerable attention for the prevention and control of metabolic disorders during pregnancy.”
XIII.
Reference 5 is not cited in the main text. Please insert an in-text citation or remove it from the reference list.
Author Response
We greatly appreciate all the Reviewers for their valuable comments and feedbacks to improve our work.
Introduction
Comment 1: I recommend a more structured flow for the Introduction: 1. Start with the importance of gestational metabolic syndrome (GMS) in relation to pregnancy and metabolic disorders. 2. Then transition to the role of iodine, emphasizing the increased requirements during pregnancy and the limitations of current recommendations. 3. Highlight the extrathyroidal functions of iodine, which open new research directions. 4. Point out the knowledge gap: limited evidence regarding serum iodine concentration (SIC), especially among pregnant women in iodine-adequate areas. 5. Conclude with a clear statement of the study objective.
R: Thanks for the suggestion from the Reviewers. The introduction has revised and followed the structures suggested by the Reviewer.
Comment 2:Line 56: “Recent studies suggested that iodine, aside from its role in the thyroid, functions as an antioxidant, immunomodulator, and differentiator in various organs and tissues.” It would be beneficial to include concrete examples for these functions.
R: Thanks for the suggestion. As required, sentences have been added in the paragraph (page 2 line 61-64) as:
“ In vitro and animal experiments showed oxidized iodine can neutralize reactive oxygen species, and trigger apoptosis and differentiation effects on diverse epithelial cells6. Moreover, iodine can be uptake and metabolized by immune cells, act as an anti-inflammatory or proinflammatory agent7. ”
Comment 3: Materials and Methods
III. The FSI and TyG indices should be briefly defined upon first mention to improve clarity. Also, the formula formatting should be improved for readability, e.g., TyG-index = ln[TG (mg/dL) × FPG (mg/dL) / 2].
R: Thanks for the suggestion. The formulas and their definitions have been explained in the main text and footnotes of tables accordingly. Revisions have been made in section 2.3 in page 4-5 line 169-177 as:
“ Framingham steatosis index (FSI) was estimated on the basis of metabolic conditions and liver function with inclusion of parameters such as age, BMI, TG, hypertension, diabetes and ALT/AST. As a surrogate marker for non-alcohol fatty liver disease (NAFLD), FSI has been validated in Asian populations18, 19. FSI formula was modified due to its application in gestational population by replacing hypertension and diabetes with GH and GDM. The detailed equation was indicated in the footnotes of Tables. Triglyceride-glucose (TyG) index is a reliable and simple surrogate for insulin resistance with substantial relevance to metabolic disorders20. TyG-index was calculated as In[TG (mg/dL)×FPG (mg/dL) /2].”
In footnotes of all the related Tables, the formulas have been modified as:
“TyG-index=In[TG(mg/dL)×Fasting glucose(mg/dL)/2];
FSI=-7.981+0.011×age(yrs)-0.146+0.173×BMI at T1(kg/m2) +0.007×TG(mg/dL) +0.593×GH(yes=1,no=0) +0.789×GDM(yes=1, no=0)+1.1×[ALT/AST≥1.33(yes=1, no=0)].”
Comment 4:The use of multiple statistical software packages is appropriate. However:
A) A more detailed description of the data transformation process would be helpful—how was normality assessed, and were any other transformations applied?
R: Thanks for the comments. Due to the relatively large sample size, data normality was tested by QQ plot. Variables of heterogenous variance were logarithmic transformed before parametric approaches. As suggested, the revisions have been made in section 2.5 (Statistical analysis, page 5 line 198-200) as: “Data normality was visually tested by QQ plot. Variables of skewed distribution or heterogenous variance (SIC, body fat% ,TG, TSH, FT3/FT4) were logarithmic transformed before parametric approaches.”.
The footnotes in Table 1 have been revised as: “For continuous variables, data were presented as mean±SD and tested by ANOVA, or median (interquartile range, P25-P75) and tested by Kruskal-Wallis. For categorial variable, data were expressed as n(%) and tested by chi-square.”.
Comment 5: B) It should be indicated which software was used for which analysis (e.g., R for restricted cubic spline, SPSS for regression models).
R: Thanks for the suggestion. We have modified the sentence in section 2.5 of Statistical analysis in page 5 line 196-197 as:
“Data entry was made by Epi-data 3.1. Data processing and regression analysis were performed by SPSS 24.0 and R 4.3.0. Storm Statistical Platform (www.medsta.cn/software) was applied for restricted cubic spline (RCS) analysis. ”.
Comment 6: C) The specific statistical tests used to assess group differences should be explicitly stated (e.g., ANOVA, Kruskal–Wallis, chi-square test).
R: Thanks for the comments. The statistical approaches have been revised and indicated in the footnotes of Table 1 as. “For continuous variables, data were presented as mean±SD and tested by ANOVA, or median (interquartile range, P25-P75) and tested by Kruskal-Wallis. For categorial variable, data were expressed as n(%) and tested by chi-square.”
Comment 7: D) Regarding the restricted cubic spline (RCS): the rationale behind choosing the 0–35–65–95 percentiles should be clarified. If the distribution was skewed or included outliers, this could influence the spline modeling. Furthermore, it would be standard to report effect sizes and confidence intervals, not only p-values < 0.1.
R: Thanks for the Reviewer’s comments. We applied the conventional percentiles of 0–35–65–95 for RCS in order to obtain good model fitness, which was also suggested by Harrell in his book of “Regression Modeling Strategies” that, “For many datasets, k=4 offers an adequate fit of the model and is a good compromise between flexibility and loss of precision…”. Furthermore, “Placing knots at fixed percentiles of data distribution can ensures enough points in each interval, and also guards against letting outliers overly influence knot placement.”.
We have made logarithmic transformation for the independent variable of SIC, which help normalizing the skewed distribution, reducing the outliers, and facilitating the spline modeling. In our data, RCS was utilized for non-linearity testing, if non-linearity was suggested, further piece-wised regression analysis would be made. There was no need to report effect sizes and confidence intervals.
Comment 8: Results
In Table 1, it is not specified which statistical tests were used to evaluate group differences. This information should be clearly stated (e.g., t-test, Mann–Whitney U, chi-square test, or ANOVA).
R: Thanks for the suggestion. The testing methods have been added in the footnotes of Table 1 as “For continuous variables, data were presented as mean±SD and tested by ANOVA, or median (interquartile range, P25-P75) and tested by Kruskal-Wallis. For categorial variable, data were expressed as n(%) and tested by chi-square.”
Comment 9: Important findings are often referred to as “data not shown,” and numerical results (e.g., β coefficients, confidence intervals, p-values) are rarely presented, especially in mediation and sensitivity analyses. Some interpretations may overstate the strength of associations derived from cross-sectional data.
R: Thanks for the comments. We are sorry for any unclear description which may bring about misunderstanding. The MANOVA results have shown in the main text although not exhibited in the table. As we reported, “Multivariate ANOVA (MANOVA) suggested SIC at T1 had significantly associations with overall metabolic profiles by Wilks' Lambda test (p<0.001) with the major significance being observed on BMI at T1 (p=0.003), TG (p=0.023), TC (p=0.041), TyG-index (p=0.001) and TyG-BMI (p=0.060).”
For the numeric results by linear regression analysis, such as, Table 1 and supplemental tables on sensitivity or subgroup analysis, we have already presented β coefficients, confidence intervals, and p-values. For the results based on general linear models and logistic regression, like other publications, we reported the estimated mean and standard error, or odds ratio (95% CI) and p values. Mediation analysis was not conducted for current data. The study was prospective not cross-sectional design. As both maternal SIC and metabolic factors were tested in the same time point of early pregnancy, we therefore admitted the limitation in discussion as (page 18 line 583-584), “First, serum iodine and metabolic markers were tested in early pregnancy. These cross-sectional data had limitation in causality inference.”
Comment 10: Discussion
VII. Section 4.2 (“Maternal iodine with metabolic components”) is overly long and includes repetitive elements. Some statements closely repeat content from the Introduction. A more concise presentation is recommended to maintain focus.
R: Thanks for the suggestion. The section 4.2 has been refined and shortened as required. Please kindly see the paragraphs in page 15-16 line 446-482 for the detailed revisions.
Comment 11: VIII. In section 4.3 (“Maternal iodine, BMI and postpartum hemorrhage”), the proposed mechanism—excessive gestational weight gain (GWG) → larger fetus → postpartum hemorrhage—is speculative. Notably, no significant association was found with birth weight in the study.
R: Thanks for the comments. Our findings on the joint effects of maternal iodine and metabolic conditions showed that, “low maternal SIC combined with GMS or high pre-pregnancy BMI had significantly increased postpartum bleeding, even after controlling for birth weight, delivery modes and thyroid hormones.”. The proposed sequence was that, low iodine- increased GWG- large baby-postpartum hemorrhage. In the regression model for postpartum hemorrhage, we have adjusted birth weight as covariate. Our previous data have reported, high serum iodine levels were associated with decreased GWG and increased risk of SGA5.
Comment 12: Section 4.5 (“Maternal iodine, lipids and gestational duration”): A) The reported decrease in gestational duration (0.7–0.9 weeks) is considered significant, yet its clinical relevance is questionable and not fully discussed.
R: Thanks for the comments. We have revised and made further discussion accordingly as shown in page 17 line 531-539:
“We further observed that, both high SIC and hyperlipidemia were associated with a significantly reduced gestational duration (0.7-0.9 week) at delivery. The magnitude of the shortened duration (5-6 days) was comparable to a meta-analysis6 (21 randomized controlled trials and 10,802 pregnant women) on dietary fish-oil supplementation reported a 5.8-day increase in gestational age, a 22% reduced risk for early preterm delivery, 23% lowered risk of LBW, and increased infantile birth weight by 51.2 g. Whether the shortened gestation duration with high SIC and high lipids in our data implicated an increased risk of preterm delivery or LBW was unknown. The findings need to be confirmed in large- scale birth cohort.”.
Comment 13:B) The oxidative stress hypothesis is reintroduced without empirical support, as no oxidative markers were measured in the study.
R: Thanks for comment. Due to the limited resources, markers related to oxidative stress and inflammation were not tested currently. However, we have stored adequate specimens in ultra-analyzer and it is highly possible to test these markers once the budget allowed. We have admitted this limitation in discussion as:
“Third, due to the resource limitation, several metabolic components of GMS were unavailable in our data including waist circumferences, and lipid profiles other than TG and TC. Insulin level and cytokines related with oxidative stress or inflammation are not tested either. In addition, data on dietary iodine-rich food intakes and the usage of iodized salt were only available in a small group of participants. Future exploration for a wide coverage of involved biomarkers and investigation of detailed dietary iodine sources could substantiate current findings and exhibit comprehensive perspectives.”
Comment 15: Conclusions
- The Conclusions section is concise, but some expressions may be overstated (e.g., “clues to pathogenesis”). It does not discuss potential clinical applicability or implications for future guidelines. Although defining optimal SIC ranges appears to be an implicit aim of the study, this is not addressed in the conclusions.
R: Thanks for the comments. As suggested, the conclusions have been revised as:
“Our prospective data in iodine adequate Chinese pregnant women showed that, high maternal SIC during early pregnancy were associated with increased metabolic risk and afterwards adverse birth outcomes. The findings emphasize the importance of individual iodine assessment during gestation especially among mothers with metabolic abnormalities in prevention of adverse birth outcomes.”.
Comment 16: Minor Issues
- In Figure 1, the labels “Low birth weight, n = 66” and “macrosomia, n = 26” are not clearly visible.
R.Thanks. We have adjusted the box to make it visible.
XII. Lines 44–46: The sentence “Identification of novel nutritional factors therefore received numerous attentions for prevention and control of metabolic disorders during pregnancy.” should be revised to: “Therefore, nutritional factors have received considerable attention for the prevention and control of metabolic disorders during pregnancy.”
R: Thanks. The sentence has been revised and highlighted in red fonds.
XIII. Reference 5 is not cited in the main text. Please insert an in-text citation or remove it from the reference list.
R: Thanks. The reference 5 has already been indicated in the original line 54.
Round 2
Reviewer 4 Report
Comments and Suggestions for Authors
The corrections are acceptable and the manuscript is suitable for publication. There are editing errors that need to be corrected.